# The Potential of PSMA as a Vascular Target in TNBC

**DOI:** 10.3390/cells12040551

**Published:** 2023-02-08

**Authors:** Amelie Heesch, Lars Ortmanns, Jochen Maurer, Elmar Stickeler, Sabri E. M. Sahnoun, Felix M. Mottaghy, Agnieszka Morgenroth

**Affiliations:** 1Department of Nuclear Medicine, University Hospital Aachen, RWTH Aachen University, 52074 Aachen, Germany; 2Department of Obstetrics and Gynecology, University Hospital Aachen, RWTH Aachen University, 52074 Aachen, Germany; 3Center for Integrated Oncology (CIO), Aachen, Bonn, Cologne, Düsseldorf (ABCD), Germany; 4Department of Radiology and Nuclear Medicine, Maastricht University Medical Center (MUMC+), 6202 Maastricht, The Netherlands

**Keywords:** triple-negative breast cancer, prostate-specific membrane antigen, angiogenesis, tumor-endothelial crosstalk

## Abstract

Recent studies proving prostate-specific membrane antigen (PSMA) expression on triple-negative breast cancer (TNBC) cells and adjacent endothelial cells suggest PSMA as a promising target for therapy of until now not-targetable cancer entities. In this study, PSMA and its isoform expression were analyzed in different TNBC cells, breast cancer stem cells (BCSCs), and tumor-associated endothelial cells. PSMA expression was detected in 91% of the investigated TNBC cell lines. The PSMA splice isoforms were predominantly found in the BCSCs. Tumor-conditioned media from two TNBC cell lines, BT-20 (high full-length PSMA expression, PSMAΔ18 expression) and Hs578T (low full-length PSMA expression, no isoform expression), showed significant pro-angiogenic effect with induction of tube formation in endothelial cells. All TNBC cell lines induced PSMA expression in human umbilical vein endothelial cells (HUVEC). Significant uptake of radiolabeled ligand [^68^Ga]Ga-PSMA was detected in BCSC1 (4.2%), corresponding to the high PSMA expression. Moreover, hypoxic conditions increased the uptake of radiolabeled ligand [^177^Lu]Lu-PSMA in MDA-MB-231 (0.4% vs. 3.4%, under hypoxia and normoxia, respectively) and MCF-10A (0.3% vs. 3.0%, under normoxia and hypoxia, respectively) significantly (*p* < 0.001). [^177^Lu]Lu-PSMA-induced apoptosis rates were highest in BT-20 and MDA-MB-231 associated endothelial cells. Together, these findings demonstrate the potential of PSMA-targeted therapy in TNBC.

## 1. Introduction

Triple-negative breast cancer (TNBC), which occurs especially in young women, is a highly aggressive breast cancer associated with a poor prognosis. The absence of progesterone and estrogen receptors and a low HER2 expression severely limit the therapeutic possibilities. The current therapy regime includes surgery, chemotherapy, and radiation [1]. TNBC initially responds well to chemotherapy, but the outcome does not correlate with overall survival, and the risk of relapse (hazard ratio, 2.6; 95% confidence interval, 2.0–3.5; *p* < 0.0001) and death (hazard ratio, 3.2; 95% confidence interval, 2.3–4.5; *p* < 0.001) remains high [2,3]. One major problem in TNBC treatment represents the breast cancer stem cell (BCSC) population which causes tumor recurrence and progression due to its self-renewing and multipotent capacities [4]. Although this is not unique to TNBC, the BCSC population is especially large in this tumor entity [5]. This makes the need for targeted therapies more urgent than ever, and the prostate-specific membrane antigen (PSMA) has recently been focused on as a promising candidate. As a membrane-bound receptor, it undergoes clathrin-mediated endocytosis. Some PSMA molecules are degraded, and others are recycled back to the plasma membrane [6]. The process of internalization occurs through the endocytic targeting motif MWNLL, the first amino acids of the cytoplasmic tail [7]. PSMA is not only expressed on the surface of prostate cancer (PCa) cells but also on other solid tumors such as breast cancer and the surrounding vasculature [8]. Interestingly, estrogen and progesterone-negative breast tumors exhibit a higher PSMA expression than receptor-positive phenotypes [9]. As recently shown, PSMA expression in PCa correlates inversely with survival and increases with tumor grade [10], highlighting the aggressiveness of PSMA-positive entities. To date, several studies have linked PSMA as an important actor in cancer-related angiogenesis. This transmembrane protein regulates endothelial cell invasion and angiogenesis by modulating integrin and p21-activated kinases (PAK) signaling in endothelial cells [11] and promotes tumor angiogenesis by generating small, pro-angiogenic laminin peptides [12]. Several studies have demonstrated that different types of cancer cells are able to induce PSMA expression in endothelial cells [13,14,15]. As suggested by these studies, PSMA contributes to the regeneration of endothelial nitric oxide synthase (eNOS), an important factor in regulating angiogenesis, by increasing local folate levels [16]. Therefore, induced PSMA expression with its enzymatic folate hydrolase in endothelial cells might correlate with increased angiogenesis [17]. Interestingly, as shown for the PCa cell line LNCaP, the released PSMA-positive membranes are able to induce the high-angiogenic state in endothelial cells [18]. Additionally, cancer cells were shown to support neovascularization by secretion of pro-angiogenic factors such as vascular endothelial growth factor (VEGF), which stimulate the proliferation of endothelial cells [19]. Currently, there are several anti-angiogenic agents, such as bevacizumab, sorafenib, and sunitinib, available. They inhibit the VEGF receptor and have been used in clinical trials for TNBC with limited efficacy [20,21]. A recent study observed that targeting the VEGF receptor with sunitinib in TNBC mice models led to the regrowth of the tumor with decreased survival after discontinuation of the treatment [22]. Based on these data, the need for the advancement of TNBC therapy by discovering new targets is obvious, and PSMA may be a suitable candidate. 

The treatment regime of PCa already utilizes PSMA as a therapeutic target in [^177^Lu]Lu-PSMA radioligand therapy (RLT). Usually, patient selection criteria include metastatic castration-resistant disease, failure or ineligibility for cytotoxic treatment, adequate organ function, and PSMA-positivity in PSMA-PET/CT [23]. Although the overall survival ranges from 12 to 17.6 months after initiating PSMA-targeted therapy in heavily pretreated PCa patients, there is still a subpopulation of patients who do not respond to the therapy despite PSMA-positivity [24]. Some key effectors, such as TP53, MYC, and the androgen receptor, were suggested to contribute to the [^177^Lu]Lu-PSMA therapy resistance [25]. The underlying mechanisms still need to be investigated. An alternative splicing generating several PSMA isoforms, some of which are localized in the cytosol, might be one of the mechanisms leading to therapy resistance [26,27,28,29]. The identified isoforms of PSMA are PSM’, PSM-C, PSM-D, PSM-E, PSMAΔ6, and PSMAΔ18. The splice mechanisms include the usage of an alternative splice site (PSM’, PSM-C, PSM-D, and PSM-E) and exon exclusion (PSMAΔ6 and PSMAΔ18) [27,28]. PSM’ lacks the intracellular and transmembrane domain resulting in a cytosolic localization, but it is still enzymatically active [26]. It represents the predominant form in benign prostate, whereas the PSMA/PSM’ ratio is increased in PCa [27]. PSM-C and PSM-D also occur cytosolic and are mainly expressed in lymph node and bone metastases [27]. PSMAΔ6 and PSMAΔ18 both harbor the transmembrane domain but are enzymatically not functional. Their exact role in PCa is still unknown, but PSMAΔ6 represents the least expressed PSMA variant [28]. PSM-E is the most recently discovered isoform which was only detected in PCa, not in other solid tumors. It is enzymatically active and is localized to the cell membrane [29]. Schmittgen et al. observed that PSMA, PSM‘, PSM-C, and PSM-D show the lowest expression in liver metastases of PCa compared to primary tumors, bone and lymph metastases, and even normal prostate tissue [27]. 

Thus, in this study, a panel of 12 TNBC cell lines was analyzed regarding the PSMA isoforms expression. Furthermore, the impact of TNBC cells on neovascularization and PSMA expression on endothelial cells was explored. These results have a high impact on breast cancer research, as they provide data on PSMA expression in a large panel of TNBC cells, BCSCs, and tumor-associated endothelial cells in vitro. It may also help to evaluate the efficacy of endogenous radiotherapy using [^177^Lu]Lu-PSMA in TNBC patients.

## 2. Materials and Methods

### 2.1. Cell Lines and Culture Conditions

The endothelial cell line HUV-EC-C (referred to as HUVEC), benign breast epithelial cell line MCF-10A, PCa cell line LNCaP, and all TNBC cell lines MDA-MB-231, MDA-MB-468, BT-20, Hs578T, SUM149PT, SUM1315MO2, and HCC1937 were purchased from ATCC (Virginia, USA). HUVEC were cultured in endothelial cell basal medium (EBM-2) supplemented with endothelial growth medium (EGM-2) SingleQuots (Lonza, Switzerland) and 10% fetal bovine serum (FBS). MDA-MB2321, MDA-MB-468, and BT-20 were cultured in Dulbecco’s Modified Eagle’s Medium (DMEM) (Pan Biotech, Germany) supplemented with 10% FBS and 1% penicillin-streptomycin. MCF-10A cells were cultured in DMEM/F12 (Thermo Fisher Scientific, USA/MA) supplemented with 5% horse serum, 20 ng/mL endothelial growth factor (EGF), 0.5 µg/mL hydrocortisone, 100 ng/mL cholera toxin, 10 µg/mL insulin, 1% penicillin-streptomycin. LNCaP and HCC1395 were cultured in Roswell Park Memorial Institute media (RPMI) (Pan Biotech, Germany) supplemented with 10% FBS and 1% penicillin-streptomycin. SUM1315MO2 were cultured in Ham’s F12 (Thermo Fisher Scientific, USA/MA) supplemented with 10 ng/mL EGF, 5% FBS, 10 mM 4-(2-hydroxyethyl)-1-piperazineethanesulfonic acid (HEPES), 1 µg/mL hydrocortisone, 5 µg/mL insulin, and 1% penicillin-streptomycin. SUM149PT were cultured in 5% FBS, 10 mM HEPES, 1 µg/mL hydrocortisone, 5 µg/mL insulin, and 1% penicillin-streptomycin. The cells were subcultured twice a week. BCSC1 and BCSC2 [30], BCSC3 and BCSC4 [31], and BCSC5 [32] were a gift from the Department of Obstetrics and Gynecology, RWTH University Hospital Aachen and were maintained in mammary epithelial cell growth basal medium (MEBM) (Lonza, Switzerland) supplemented with 1% B27, 1% amphotericin, 1% penicillin-streptomycin, 20 ng/mL EGF, 4 μg/mL heparin, 20 ng/mL fibroblast growth factor (FGF), 35 μg/mL gentamicin, and 500 nmol/L rho kinase inhibitor. BCSCs were subcultured once a week and maintained in culture media containing 2% growth-factor reduced Matrigel (Corning, USA/NY). All cells were maintained at 37 °C and 5% CO_2_ except BCSCs under 3% O_2_/5% CO_2_. The tube formation assay was carried out in serum- and additive-free DMEM. All cells were tested biweekly for mycoplasma contamination.

### 2.2. Radiochemistry

Both radiopharmaceuticals, [^68^Ga]Ga-PSMA-11 and [^177^Lu]Lu-PSMA-I&T, were produced by a routine procedure primarily used for patient care. For this purpose, a cassette synthesizer-type GRP 3 V (Scintomics, Germany) was used with cassettes (ABX, Germany) dedicated to the two radionuclides (SC-01 for ^68^Ga using HEPES buffer and SC-05 for ^177^Lu using acetate buffer during labeling). Briefly, 10 mL of ^68^GaCl3 containing generator eluate (0.6 M) with a radioactivity of up to 1.5 GBq was diluted with water and trapped on a cation exchange SPE cartridge, eluted with 5 M NaCl, and added to the reactor containing precursor PSMA-11 and HEPES. After labeling reaction (120 °C, 10 min), HEPES was removed using reversed phase SPE extraction. The product was eluted with aqueous EtOH, followed by formulation with PBS. Up to 2 GBq (< 1mL) ^177^LuCl3 (ITM, Germany) was transferred to the reactor containing precursor PSMA-I&T and acetate buffer. After 20 min reaction at 100 °C, the solution was quenched with a saline solution containing DTPA. Radiochemical purities and yields were > 95% (measured with radio-HPLC).

### 2.3. RNA Isolation and Reverse Transcription

Total RNA was isolated from all cell lines using RNeasy Plus Mini Kit (Qiagen, Germany) according to the manufacturer’s instructions. cDNA was synthesized with the Advantage RT-for-PCR Kit (Takara, Japan) using oligo (dT) primer.

### 2.4. PCR

Human PSMA/isoform cDNA was amplified with the following PCR primer pairs (0.5 µM): PSMA: 5‘-TGCAGGGCTGATAAGCCAGGCATT-3‘ and 5’-TGGGATTGAATTTGCTTTGCAAGCTG-3’; PSM’: 5’-TGGACCCCAGGGTGGTTTAT-3’ and 5’-GCATCCCAGCTTGCAAACAA-3’; PSM-C: 5’-GGTACTGATTTGCAGACTTGATCC-3’ and 5’-GCTTTGGAGTAATGTTAGTAGCTTCAT-3’;PSM-D: 5’-CCAGGCAGGTTAAAAGCCAA-3’ and 5’-CCTCTCTGCCAGACACCCAG-3’PSM-E: 5’-GAATCTCCTTCACGAAACCG-3’ and 5’-ATAAACCACCCCAGCCTCTC-3’; PSMΔ6: 5’-AATTGGAACGGGACATGAA-3’ and 5’- CCTCTGCAATTCCACGCC-3’; PSMΔ18: 5’- GAAACAAACAAATTCAGCGG-3‘ and 5’-AGAGCATCATAAATTCCTGG-3’; human β-actin: 5’-AGAGCTACGAGCTGCCTGAC-3’ and 5’-CACCTTCACCGTTCCAGTTT-3’. Samples were diluted with 6 × loading dye and loaded on a 1% agarose/ethidiumbromide gel. A voltage of 100 V was applied for 1–2 h. Afterwards, the signal was detected with an ImageQuant LAS 4000 Luminescent Analyzer (GE Healthcare, USA/IL).

### 2.5. qPCR

Human PSMA/isoform cDNA was amplified with the following qPCR primer pair (0.5 µM): PSMA: 5‘-GGAGAGGAAGTCTCAAAGTGCC-3’ and 5’-TGGTTCCACTGCTCCTCTGAGA-3’; human β-actin: 5’-AGAGCTACGAGCTGCCTGAC-3’ and 5’-CGTGGATGCCACAGGACT-3’. Samples were analyzed with a LightCycler 480 II (Roche, Switzerland).

### 2.6. Western Blot Analysis

Crude cell lysates were prepared from cells using 100 µL RIPA buffer, 2 µL phenylmethylsulfonyl fluoride (1 mM), and 100 µL protease inhibitor cocktail (stock: 1 tablet/5 mL water). Protein concentrations were determined using the Pierce BCA Protein Assay Kit (Thermo Fisher Scientific, USA/MA). A total of 30 µg of protein was loaded in each well on an Any kD™ Mini-PROTEAN® TGX™ Precast Protein Gel (Bio-Rad, USA/CA) and blotted onto a polyvinylidene fluoride (PVDF) membrane. Unspecific signal was blocked with 5% milk powder in TBST. The membranes were incubated overnight at 4 °C with primary antibodies: mouse α-PSMA (Abcam, 1:1000) or rabbit α-GAPDH (Cell Signaling, 1:2000). After washing, the secondary antibody horse α-mouse IgG HRP (Cell Signaling, 1:1000) or goat α-rabbit IgG HRP (Cell Signaling, 1:1000) was applied for 1 h at room temperature. The membrane was prepared for signal detection with Pierce™ ECL Western Blotting Substrate (Thermo Fisher Scientific, USA/MA) and subsequently analyzed with an Image Quant LAS 4000 Luminescent Analyzer (GE Healthcare, USA/IL).

### 2.7. Co-Cultures

HUVEC were seeded with 2.5 × 10^4^ per well on coverslips in a 24- transwell co-culture plate. After 24 h, the media was changed, and an insert with a membrane diameter of 0.45 µm with 2.5 × 10^4^ TNBC cells was placed in the wells. The co-culture was incubated at 37 °C/5% CO_2_ for an additional 96 h.

### 2.8. Immunocytochemistry

The inserts were discarded, and HUVECs were washed with PBS, fixed with 4% PFA for 15 min, and permeabilized with 0.1% triton/PBS. Unspecific signal was blocked with 5% goat serum. Samples were stained overnight at 4 °C with primary rabbit α-cluster of differentiation 31 (CD31) (Invitrogen, 1:100) and mouse α-PSMA (Abcam, UK, 1:250) antibodies. As secondary antibodies, a goat α-mouse dylight 488 (Abcam, UK, 1:1000) and a goat α-rabbit AF 555 (Cell Signaling Technology, USA/MA, 1:1000) were applied for 1 h at room temperature. DAPI staining was applied for 3 min, and the cells were mounted with Mowiol. Images were acquired with a Zeiss Imager Z.1 microscope (Zeiss, Germany).

### 2.9. Generation of Tumor Conditioned Media (TCM)

Cells at 80–90% confluence were washed 2 × with PBS. Serum- and additive-free DMEM was added to the cells and incubated for 48 h. The media was centrifuged at 1500 rpm for 5 min and the supernatant was stored at −20 °C until use.

### 2.10. Tube Formation Assay

HUVECs (3 × 10^4^ cells/well) were plated on 100 µL growth-factor reduced matrigel in 48-well plates in 200 µL TCM for 5 h at 37 °C/5% CO_2_. Tube formation was observed and imaged with a Motic AE31E microscope (Motic, China). For quantification, total tube length and total number of tubes were determined using the ImageJ macro ‘Angiogenic Analyzer’ (ImageJ version 1.53t).

### 2.11. Spheroid Formation

Equal volumes of tumor and endothelial cells (2 × 10³ + 2 × 10³) were seeded in 100 µL culture media in agarose-coated 96 wells. After 24 h, 100 µL of culture media containing 10% matrigel was added. The spheroids were allowed to grow for 7 days. For hypoxic conditions, spheroid plates were incubated in a hypoxia chamber (Billups Rothenberg, USA/CA) containing 0.5% O_2_. Every 3 days, the chamber was gassed for 3 min.

### 2.12. Immunohistochemistry

Spheroids were fixed with 4% PFA for 15 min, embedded in Tissue-Tek O.C.T. (Sakura Finetek, Germany), and snap-frozen in cold 2-methylbutan (Carl Roth, Germany). Cryosections (10 µm) were obtained using a Leica JUNG CM3000 cryostate (Leica Microsystems, Germany) and transferred on SuperFrost slides (Thermo Fisher Scientific, USA/MA). Sections were immersed in PBS, permeabilized with 0.1% triton/PBS, and blocked with 5% goat serum. Samples were stained overnight at 4 °C with primary rabbit α-CD31 (Invitrogen, USA/MA, 1:100) and mouse α-PSMA (Abcam, UK, 1:250) antibodies. After washing, sections were incubated with α-mouse Alexa Fluor 488 (Thermo Fisher Scientific, USA/MA, 1:1000) and goat α-rabbit Alexa Fluor 555 (Cell Signaling, USA/MA, 1:1000) antibodies for 1 h at room temperature. DAPI staining was applied for 3 min and the cells were mounted with Mowiol. Images were acquired with a Zeiss Imager Z.1 microscope (Zeiss, Germany).

### 2.13. Cell Uptake Experiments

Cells (1 × 10^5^ cells/well) were seeded in triplicates in 12-well plates 24 h prior to incubation with [^68^Ga]Ga-PSMA or [^177^Lu]Lu-PSMA (1 MBq/well). After 1 or 4 h, cells were washed and detached with trypsin. Cell-associated radioactivity was measured with a Wizard2 gamma counter (PerkinElmer, USA/MA). For the hypoxic conditions, the well plate was incubated for 7 days under 0.5% O_2_.

### 2.14. Flow Cytometry

Cells (1 × 10^6^ cells/well) were seeded in triplicates in 6-well plates. After 24 h, [^177^Lu]Lu-PSMA (10 MBq/well) was applied for another 24 h. Cells were thoroughly washed and fresh media was applied. Cells were post-incubated for 5 days. The amount of apoptosis after treatment was evaluated using an Annexin V-FLUOS staining kit (Roche, Switzerland), according to the manufacturer’s instructions. Stained cells were analyzed using a CytoFLEX B2-R0-V2 flow cytometer (Beckman Coulter, USA/CA). For the untreated controls, DMEM without the tracer was applied.

### 2.15. Quantification and Statistics

All quantitative experiments were performed three times in triplicates. For data analysis, ImageJ (Wayne Rasband, USA/MD, version 1.53t) and GraphPad PRISM (GraphPad Software, USA/CA, version 8.0.1) were used. Statistical analysis was performed correspondingly with one- or two-way ANOVA, and multiple comparisons were corrected via Tukey.

## 3. Results

### 3.1. PSMA is Expressed in TNBC Cells and BCSCs

To investigate the potential of PSMA-targeting in TNBC, the expression of PSMA and its isoforms at mRNA (Figure 1A) and protein level (Figure 1D,E) was analyzed in a panel consisting of 12 different TNBC cell lines. To our knowledge, this is the first large-scale PSMA expression study in TNBC cell lines. The PSMA expression at the mRNA level in the endothelial cells was investigated after co-culture with TNBC cells for 96 h (Figure 1B,C). The PCa cell line LNCaP served as positive control and the benign breast epithelial cells MCF-10A cell line as negative control. The full-length PSMA (FL-PSMA) transcript (424 bp) was expressed in all TNBC lines except HCC1937. The highest expression was detected in MDA-MB-231, BT-20 and BCSC5, BCSC3, BCSC1, and BCSC2. Among the investigated isoforms, the highest expression was visualized for the PSMA∆18 in MDA-MB-231, BT-20, SUM1315MO2, and in the FL-PSMA negative HCC1937 cell line. Interestingly, the stem cells expressed 4 of 6 investigated isoforms, with PSMA∆18, PSM-D, and PSM-E being the most prominent ones. The Western blot analysis revealed two bands at ~50 and ~100 kDa, corresponding to isoforms/degradation products and the full-length PSMA, respectively. The HUVEC co-culture experiments indicate the highest induction of PSMA expression by MDA-MB-231 and BT-20 (~3-fold gene expression) compared to MCF-10A cells (Figure 1B,C). This effect was weaker in a co-culture with the PCa line LNCaP. Interestingly, for the BCSC co-cultures, no to low induction of PSMA expression was detected.

### 3.2. TNBC Cells Promote Tube Formation In Vitro

Since PSMA is involved in cancer-related angiogenesis [11,12], it was tested if TNBC cells are able to induce an angiogenic state in HUVECs (Figure 2). Phase contrast images are shown in Appendix A. The complete EGM-2 growth media served as positive control. Considering the total number (Figure 2A) and length of generated tubes (Figure 2B), a significant tube formation was detected for BT-20 (number: *p* = 0.002; length: *p* = 0.002) and Hs578T cells (number: *p* = 0.002; length: *p* = 0.04). This pro-angiogenic effect was significantly lower for BCSCs and benign MCF-10A cells. In general, the master segment length strongly correlated with the master segment number.

### 3.3. TNBC Cells Induce PSMA Expression in HUVECs and Affect Their Morphology

The endothelial PSMA expression was investigated on HUVEC cells after 96 h of co-culture with TNBC cells (Figure 3). As controls, HUVEC were incubated either as mono-culture (Figure 3a) or with MCF-10A cells (Figure 3b). The secondary antibody control is shown in Appendix A. PSMA expression (green) was virtually absent in both controls; only the endothelial marker CD31 (red) was detected. The co-culture of TNBC cells with HUVEC strongly induced endothelial PSMA expression. Interestingly, the cells changed their morphology from the typical cobblestone appearance to an elongated one. This effect was observed solely in solitary but not in grouped growing cells. The HUVECs frequently developed direct cell–cell connections via elongated protrusions (Figure 3t–v). For all co-cultures, PSMA was detected in the cytosol and on the cell membrane, as indicated by the co-localization with CD31 protein for the cell surface expression. Interestingly, the expression of endothelial marker CD31 was higher in cells showing a PSMA signal. 

### 3.4. PSMA is Expressed in Endothelial Cells in TNBC/HUVEC Spheroids

Additionally, the PSMA induction in HUVECs was explored in a 3D tumor spheroid model consisting of TNBC cells and HUVECs (1:1) (Figure 4). The PSMA expression was analyzed after 7 days of incubation in a normoxic (20% O_2_) or hypoxic (0.5% O_2_) environment (Figure 4A). 

The spheroids were investigated for distribution of HUVECs and TNBC cells and the PSMA expression (Figure 4B,C). In the spheroids with MDA-MB-231 cells, HUVECs were preferentially located in the core region under normoxic as well as hypoxic conditions. In the MCF-10A spheroid, the HUVEC were mostly absent. PSMA expression was, in general, higher in spheroids growing under hypoxic conditions, especially in the core region. Importantly, the hypoxia increased the fraction of PSMA-expressing endothelial cells. BT-20 spheroids demonstrated the highest upregulation of PSMA under hypoxia, not only on the endothelial cells but also on the tumor cells. Co-localization of PSMA and CD31 under normoxia was observed in all spheroids except in those with MCF-10A. Under hypoxia, the MCF-10A showed a diffuse PSMA signal in the core region.

### 3.5. Radiolabeled PSMA-Ligands are Taken up by TNBC Cells

The uptake of [^68^Ga]Ga-PSMA was investigated in TNBC cell lines (Figure 5A). The highest cell uptake was detected in BCSC1 and BCSC2, with 4.2% and 2.8% after 4 h, respectively. TNBC cell lines, in general, showed lower binding and uptake of [^68^Ga]Ga-PSMA.

Hypoxia significantly increased binding and uptake of [^177^Lu]Lu-PSMA in MCF-10A (0.3% vs. 3%) and MDA-MB-231 (0.4% vs. 3.4%), whereas the PCa cells LNCaP showed decreased uptake under hypoxia (19% vs. 6.7%) (Figure 5B). In the co-culture experiments, HUVECs co-cultured with BT-20 showed the highest [^177^Lu]Lu-PSMA uptake (1%). The PSMA uptake of the HUVECs co-cultured with MDA-MB-231 and BT-20 (0.6% and 1%) was higher than the uptake of MDA-MB-231 and BT-20 cells alone (0.3% and 0.6%).

### 3.6. [^177^Lu]Lu-PSMA Induces Apoptosis in TNBC-Associated HUVECs

As PSMA is proposed as a potential target for TNBC treatment, it was evaluated whether [^177^Lu]Lu-PSMA has an apoptotic effect on the TNBC cells and the associated endothelial cells. Therefore, 10 MBq [^177^Lu]Lu-PSMA was applied and removed after 24 h. After 5 days, cells were imaged with a phase contrast microscope (Appendix A), stained with Annexin-V and Propidium Iodide for detection of apoptosis, and quantified using flow cytometry (Figure 6). The cells in the control condition were treated with DMEM (‘untreated’). The amount of apoptosis for the treated and untreated samples was measured after 5 days. A small amount of apoptosis is also common for untreated samples (not induced by [^177^Lu]Lu-PSMA); therefore, this unspecific apoptotic rate was subtracted from the apoptotic rates of the treated samples to exclude the amount of non-[^177^Lu]Lu-PSMA-induced apoptosis.

The treated TNBC cell lines MDA-MB-231 and BT-20 showed a higher apoptotic rate than MCF-10A. [^177^Lu]Lu-PSMA had no effect on the MCF-10A control. In the HUVEC, the highest apoptosis was detected in co-cultures with TNBC cell lines.

## 4. Discussion

Recently PSMA has drawn attention as a target that is expressed in TNBC cells and their associated vasculature [14]. PSMA acts as glutamate carboxypeptidase with folate hydrolase [33] and NAALADase activities [34]. Until now, PSMA expression was mostly analyzed in PCa. In this study, we report PSMA expression in a panel of 12 TNBC cell lines, including BCSCs. PSMA transcripts were detected in all cell lines except HCC1937. The strongest PSMA expression was observed in MDA-MB-231 and BT-20. Among the investigated cell lines, the BCSCs showed the highest expression of almost all isoforms of PSMA, with the exception of PSM-C and PSMAΔ6. In particular, two splice variants, PSM-E and PSM-D, were detected in all investigated BCSCs. As shown for PCa, PSM-E expression strongly correlates with tumor grade and, therefore, may serve as a prognostic marker in diagnostics [29]. For PSM-D, the expression was found to be 2-fold higher in lymph node and bone metastases of PCa patients compared to the primary tumors [27]. Importantly, the lowest expression of PSMA, PSM‘, PSM-C, and PSM-D was detected in liver metastases of PCa. Seifert et al. found that the efficiency of [^177^Lu]Lu-PSMA therapy is limited in PCa patients with liver metastases. They hypothesized that the neuroendocrine transdifferentiation of PCa cells could be responsible for this [35]. Characterization of tissue distribution of PSMA RNA demonstrated further that the expression at 2.8 kb in the liver is weaker compared to other organs, but there is also the expression of a smaller variant at 1.5 kb [36]. They suggested that it may be either a splice isoform or degraded PSMA. In liver cancer, decreased levels of folate, which is a potential metabolite of PSMA, are associated with worse survival [37]. Moreover, PSMA expression increases folate uptake [38], therefore it may be possible that in liver metastases of PCa patients, PSMA isoforms with cytosolic localization and no enzymatic function are expressed, leading to low folate levels and, therefore, worse survival of the patients. Together, this could contribute to the inefficiency of the [^177^Lu]Lu-PSMA therapy in patients with liver metastases. As some of the expressed PSMA isoforms are located in the cytosol, they would simply escape the cell membrane-targeting radioligand. With the use of drug delivery systems such as nanogels, it would be possible to also target the intracellular isoforms. In TNBC, the prognosis is especially poor due to the cancer stem cell population. As shown in this study, 80% of investigated BCSCs showed an increased PSMA expression. Cancer stem cells are generally considered very challenging due to their potential to initiate cancer growth, generate metastasis, and support disease recurrence. Moreover, they are linked to therapy resistance, and the BCSC population is particularly large in TNBC compared to other types of breast cancers [5]. Thus, the expression pattern of PSMA, including cell membrane and cytosolic localized enzymatically inactive isoforms, might contribute to therapy resistance. A common BCSC marker is CD44, a receptor for hyaluronic acid involved in tumorigenesis, invasion, and metastasis, as well as therapy response. By interacting with the Ras-Raf-Mek-Erk-cyclin D1 and the P13K pathway, it accelerates proliferation and suppresses apoptosis [39]. The BCSCs in this study are all CD44^+^ [31]. Moreover, CD44 is not exclusively expressed in cancer stem cells (CSCs) but also in claudin-low breast cancer cell lines such as MDA-MB-231 and Hs578T [40]. As a promoter for cysteine uptake and thus GSH, CD44v contributes to an increased ability to defend reactive oxygen species (ROS) [41]. An increase in ROS level occurs under hypoxic conditions, e.g., in the tumor core [42] and as a result of chemotherapy and radiation [43]. To keep the ROS level low and, therefore, prevent DNA damage and cell death, BCSCs increase the synthesis of glutathione, which is accompanied by an upregulated cysteine uptake [44]. PSMA, with its enzymatic activity as glutamate carboxypeptidase II, seems to contribute under hypoxic conditions to the ROS-scavenger mechanism by providing glutamate. Its upregulated expression induced by hypoxia, as detected here, suggests a similar role to CD44 during adaptation to an environment with low oxygen levels. 

The underlying force promoting metastasis is angiogenesis. The newly formed vessels not only serve as oxygen and nutrient supply but also facilitate the dissemination of cancer cells [45]. Thus, the impact of TNBC cells on the behavior of endothelial cells was investigated. The MDA-MB-231 induced PSMA to the highest amount, followed by BT-20. As shown by Nguyen et al., the induction of PSMA expression in HUVEC was notably lower in the co-culture with PSMA-positive LNCaP cells [13]. The viability of the HUVECs co-cultured with TNBC cells after [^177^Lu]Lu-PSMA treatment confirmed these results. As shown for PCa, androgen signaling plays a major role during pathogenesis [46]. The androgen-independent LNCaP-SF was able to induce a significantly higher amount of angiogenic markers in endothelial cells than the highly androgen-expressing parental LNCaP cell line [47]. MDA-MB-231 and BT-20 express very low levels of the androgen receptor (AR) [48]. This suggests that the level of AR expression inversely correlates with the induction of angiogenic markers and PSMA in endothelial cells.

Further, the angiogenic potential of the different TNBC lines was assessed using a tube formation assay. Incubation in BT-20 and Hs578T TCM induced the highest formation of endothelial tubes. Both cell lines originate from the basal-like subtype and share different similarities. One of them is the deletion of p16, which is a protein closely related to clinical outcomes in TNBC. The presence of p16 is linked to a good response to chemotherapy [49]. Moreover, p16 is also linked to angiogenesis. The restoration of wildtype-p16 was found to inhibit angiogenesis in human gliomas [50]. A different study demonstrated that overexpression of p16 suppresses tumor cell invasion by reducing matrix metalloproteinase 2 (MMP-2) [51]. Interestingly, MMP2 was also downregulated in LNCaP when PSMA expression was blocked, indicating that PSMA functions downstream of MMP-2 [52]. Thus, it might be possible that the high angiogenic potential and induced PSMA expression are actually related to the p16 deletion in these cell lines. As shown in PSMA knockout TRAMP mouse models, the PSMA-negative tumors are smaller, lower-grade, and more apoptotic with fewer vessels than the PSMA-positive ones [53]. This indicates that PSMA expression is closely connected to enhanced angiogenesis and, thus, the aggressiveness of the tumor. In contrast, TCM from BCSC1 and BCSC2 was not able to induce tube formation in HUVECs. This agrees with a recently published study that demonstrated that TCM from BCSC1 does not induce network formation in HUVEC, but if stimulated with tumor necrose factor-α (TNF-α), the pro-angiogenic potential increases significantly [54]. Despite similar PSMA expression, MDA-MB-231 showed a lower pro-angiogenic potential than BT-20. The TNBC secretome consists of cytokines, growth factors, different proteins, enzymes, and peptide hormones [55]. Therefore, BT-20 probably releases other or more pro-angiogenic factors than MDA-MB-231.

Previous studies have shown that survival-, migration-, and angiogenesis-related genes are upregulated in HUVEC after co-culture with CL1-5 cells. Moreover, the HUVEC phenotype changed to a mesenchymal-like morphology [56]. Similarly, in this study, the TNBC co-cultured PSMA-expressing HUVECs changed their morphology. The typical cobblestone appearance switched to an elongated form with lamellopodia-like protrusions aiming to form new cell contacts and tubes. This suggests that PSMA also plays a major role in cytoskeletal dynamics needed during cell invasion. It was previously shown that PSMA is internalized via α-tubulin [57] and interacts with filamin [58]. In the Hs578T co-culture, HUVECs showed high PSMA expression with strong co-localization with the cytoskeleton. It is noteworthy that the Hs578T cells themselves showed only moderate PSMA expression. As indicated by the co-staining, PSMA-expressing HUVECs showed an increased CD31 signal. CD31 is an angiogenic marker, which also plays a major role in metastasis and is increased in brain metastases of non-small cell lung cancer [59]. In colorectal cancer, it contributes to peritoneal metastasis through epithelial–mesenchymal transition [60]. In endothelial cells, CD31 co-localizes with integrin avβ3 and is supposed to associate via cis-interactions [61]. On the other hand, PSMA and filamin interact in endothelial cells, and PAK activation disrupts this complex and downregulates PSMA activity [11]. As filamin is one of the main regulators of integrin [62], it is possible that PSMA and CD31 participate in similar mechanisms to regulate angiogenesis via integrin signaling.

The 3D spheroid model additionally confirmed the effect of hypoxia on PSMA expression. The strongest upregulation of PSMA was visualized in the BT-20/HUVEC spheroid. Importantly, this effect was detected in cancer cells and in the endothelial spheroid compartments. Correspondingly, the binding of [^177^Lu]Lu-PSMA increased significantly under hypoxia. Interestingly, PSMA-uptake under hypoxia was downregulated in LNCaP. This cell line cannot tolerate hypoxia well, and after 72 h, the surviving fraction of cells is strongly decreased [63]. MDA-MB-231, on the other hand, adapt very well to hypoxia, the apoptotic cell population decreases under hypoxia, and they start to express stem cell surface markers [64].

The therapeutic effect [^177^Lu]Lu-PSMA treatment became obvious in the viability assay, where apoptotic rates of the HUVECs were significantly higher in the TNBC co-cultures compared to the monocultures. This is not surprising as PSMA is proposed as a vascular target and is found especially on tumor-associated vasculature. Tolkach et al. found that PSMA is expressed in only 3% (n = 315) of tumor cells in breast cancer but in 60% (n = 189) of the tumor vessels [65].

## 5. Conclusions

To date, there are no ideal targeted therapeutics for TNBC patients. PSMA identification was a huge step in the development of targeted therapies in the fight against PCa. This in vitro study gives novel insights into PSMA expression and induction on a cellular level and highlights the potential of PSMA as a target for diagnosis and therapy in TNBC. Its specific expression, especially on the tumor-associated vasculature, makes it a promising candidate to target tumor angiogenesis. It is clearly demonstrated that TNBC cells strongly induce PSMA in tumor-associated endothelial cells, whereas the expression of PSMA on the TNBC cells alone is most likely not sufficient for [^177^Lu]Lu-PSMA therapy. Interestingly, hypoxia strongly increases PSMA expression in breast cancer cells, which might be addressed by fractionated [^177^Lu]Lu-PSMA therapy, with the first fraction targeting PSMA-expressing endothelial cells and, in this way, decreasing the oxygen supply. Fractionated therapy means that instead of one high dose, several lower doses of treatment are applied. Thus, PSMA represents an attractive alternative to conventional treatment strategies. 

## Figures and Tables

**Figure 1 cells-12-00551-f001:**
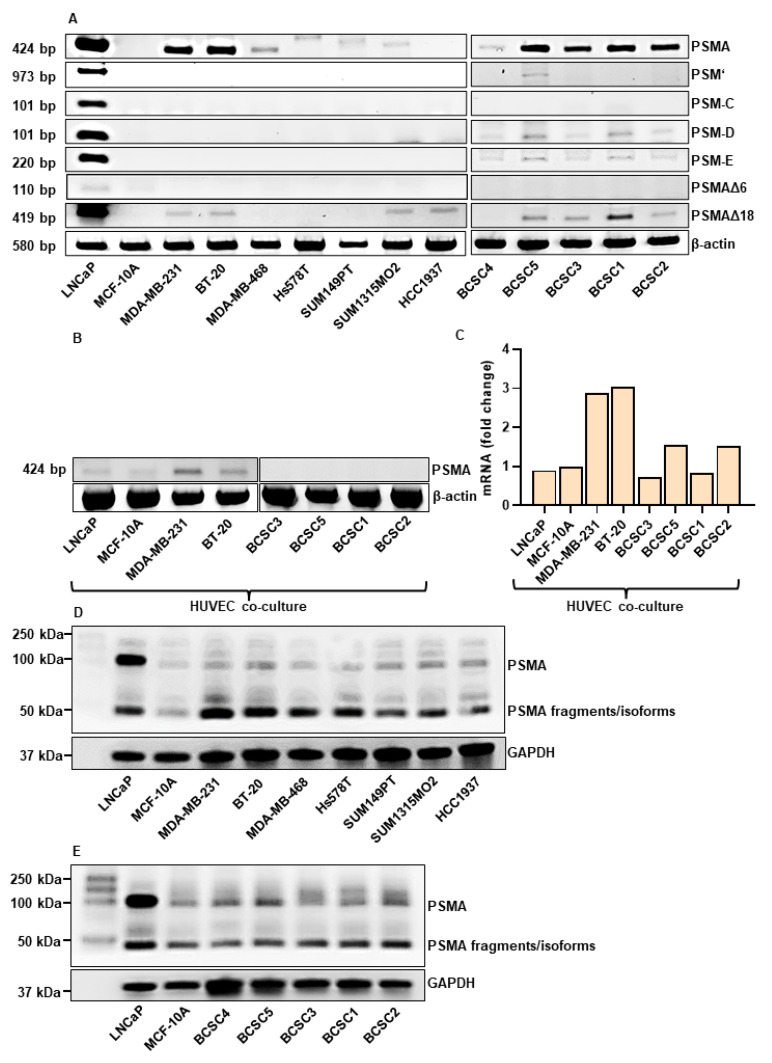
Expression of PSMA isoforms in TNBC. (**A**) RT-PCR analysis of PSMA and isoforms (PSMA, PSM’, PSM-C, PSM-D, PSM-E, PSMAΔ6, and PSMAΔ18) in the indicated TNBC cell lines and BCSCs. β-actin represents the loading control. (**B**) RT-PCR analysis of PSMA in HUVECs co-cultured for 96 h in a trans-well system with the indicated cell lines. (**C)** qPCR analysis of PSMA in HUVECs co-cultured for 96 h in a trans-well system with the indicated cell lines. (**D**,**E**) SDS/Western Blot analysis of PSMA in TNBC (**D**) and BCSC (**E**) cell lines. Total cell lysates were used. GAPDH represents the loading control.

**Figure 2 cells-12-00551-f002:**
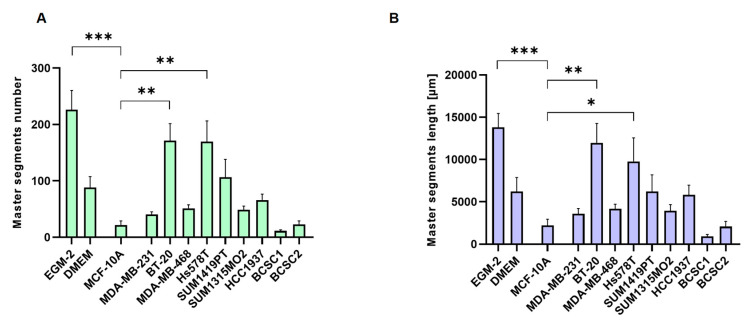
Tube formation assay with HUVEC. (**A,B**) Relative master segment number (**A**) and master segment length (**B**) were quantified using the ImageJ plugin ‘Angiogenic Analyzer’. Data are presented as mean ± SEM (n = 3). *** *p* ≤ 0.001; ** *p* ≤ 0.01; * *p* ≤ 0.05; n.s. *p* > 0.05.

**Figure 3 cells-12-00551-f003:**
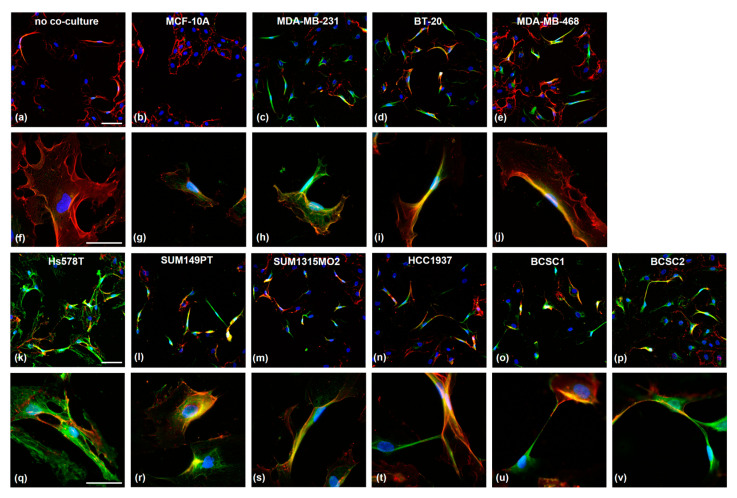
Microscopic analysis of co-cultured HUVECs after co-staining with α-PSMA (green) and α-CD31 (red) antibodies. HUVECs were cultured alone (**a**,**f**) or co-cultured with the cell lines MCF-10A (**b,g**), MDA-MB-231 (**c,h**), BT-20 (**d,i**), MDA-MB-468 (**e,j**), Hs578T (**k,q**), SUM149PT (**l,r**), SUM1315MO2 (**m,s**), HCC1937 (**n,t**), BCSC1 (**o,u**), and BCSC2 (**p,v**) for 96 h in a trans-well system (0.4 µm pore size). Nuclei were stained with DAPI. Scale bars a–e and k–p: 100 µm; scale bars f–j and q–v: 50 µm.

**Figure 4 cells-12-00551-f004:**
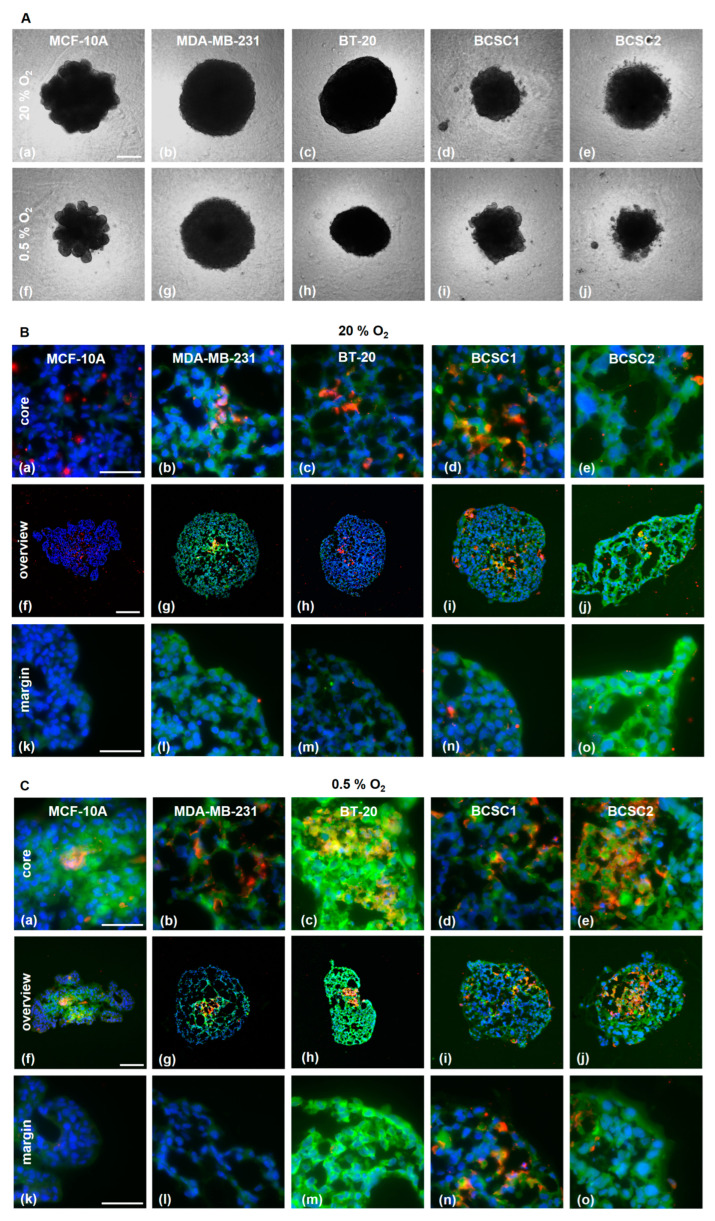
TNBC/HUVEC spheroid formation. (**A**) TNBC and HUVECs were co-seeded in equal amounts on agarose-coated wells. For 7 days, the spheroids were allowed to grow under normoxia (a–e) or hypoxia (f–j) and then imaged with a phase contrast microscope. Scale bar: 200 µm. (**B,C**) Staining with α-CD31 (red) and α-PSMA (green) antibodies of cryosections obtained from spheroids grown under normoxia (**B**) and hypoxia (**C**). Scale bars overview: (**B**) f–h; (**C**) g,h: 200 µm, (**B**) i,j; (**C**) f, i, j: 100 µm. Scale bar core and margin: 50 μm.

**Figure 5 cells-12-00551-f005:**
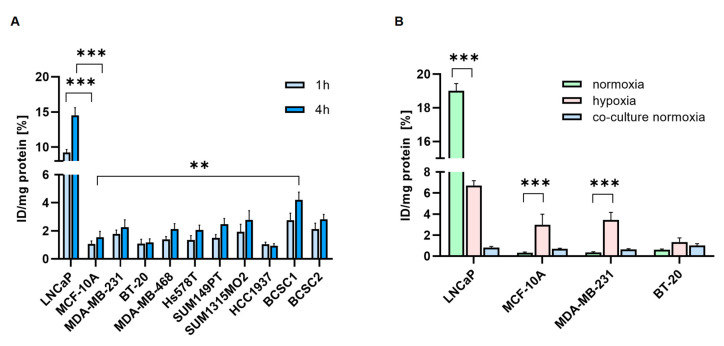
Cell uptake experiments. (**A**). Cell-associated radioactivity of [^68^Ga]Ga-PSMA was measured after 1 h (light blue) and 4 h (dark blue). (**B**). Cell-associated [^177^Lu]Lu-PSMA was measured in TNBC cells after 4 h. Cells were cultured before under normoxic (green) or hypoxic (pink) conditions. Additionally, HUVEC co-cultured for 96 h with TNBC under normoxia were measured after 4h (blue). Data are presented as mean ± SEM (n = 3). *** *p* ≤ 0.001; ** *p* ≤ 0.01; n.s. *p* > 0.05.

**Figure 6 cells-12-00551-f006:**
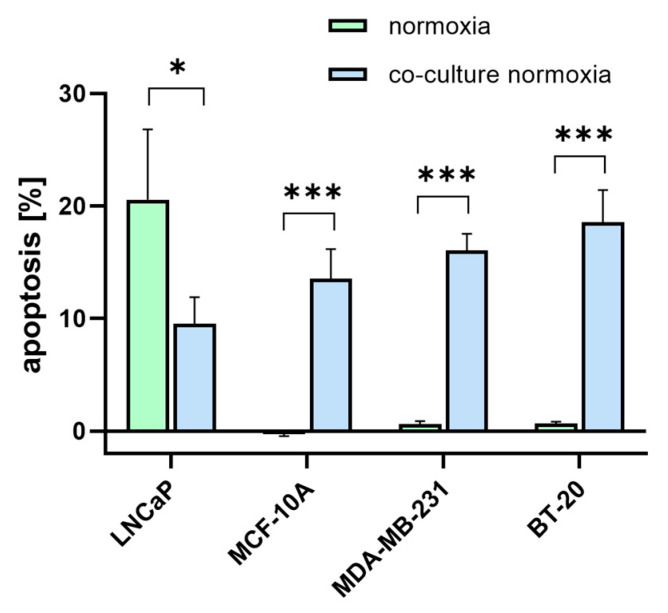
Annexin-V apoptosis assay using flow cytometry. TNBC cells (green) and HUVECs co-cultured with TNBC (blue) were treated with 10 MBq [^177^Lu]Lu-PSMA for 24 h. The amount of apoptosis was measured after 5 days post-incubation in treated and untreated samples. Untreated apoptotic rates were subtracted from treated apoptotic rates, and the differences are presented as mean ± SEM (n = 3). *** *p* ≤ 0.001; * *p* ≤ 0.05.

## Data Availability

The data presented in this study are available on request from the corresponding author.

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
