# Peer review of "The Potential of PSMA as a Vascular Target in TNBC"

_cells, 2023, doi:10.3390/cells12040551_

Round 1
Reviewer 1 Report
*How do the authors interpret the fact that the majority of cells studied in co-culture with HUVECs seems to inhibit tube formation (as compared to DMEM)?
*The claim that PSMA expression is induced in HUVEC co-cultures and spheroids seems to be supported by immunofluorescence staining only. Specificity of antibodies may vary in different settings and, in general, more than one method should be used to support a particular hypothesis. Therefore, the authors need to test for PSMA expression in co-cultures and spheroids by an additional assay of high specificity, preferably RT-qPCR of mRNA from these specimens.
*For Fig. 6, please describe in detail how cells in the control condition were treated. Which condition is compared with the other, whicht subtracted from which?
Reviewer 2 Report
It is very gratifying to see how an approved PSMA agent can be potentially used to treat a difficult disease such as TNBC. I have only minor specific comments.
1. P. 1, line 23: Please spell out HUVEC.
2. P. 1, line 46: Comment on the internalization of certain PSMA molecules.
3. P. 3, line 145: What was used to measure radiochemical purities and yields?
4. P. 5, line 222: How were 1 and 4 hr chosen for the experiment?
5. P. 8, Figure 2: MDA-MB-231 exhibited similar PSMA characteristics as BT-20 in Figure 1, what is the reason for not showing pro angiogenic effect as BT-20 in Figure 2?
6. P. 11, line 351: How was 10 MBq chosen for the experiments?
7. P. 11, line 351: What was the reason for waiting for 5 days?
8. P. 12, line 369: Any speculation on why HCC1937 was different from other TNBC cell lines?
9. P. 12, line 374: Can a ligand specifically target PSM-E expression?
10. P. 12, line 390: Can some of the expressed PSMA isoforms be capable of membrane internalization? Can a ligand be used to cross the membrane and then specifically target the expressed PSMA isoforms located in the cytosol? What are the possible reasons for some of these expressed PSMA isoforms situated in the cytosol?
11. P. 14, line 490: What would a fractionated Lu-177 PSMA therapy be like?
Round 2
Reviewer 1 Report
The manuscript is now qualified for publication.